# Is Cardiac Transplantation Still a Contraindication in Patients with Muscular Dystrophy-Related End-Stage Dilated Cardiomyopathy? A Systematic Review

**DOI:** 10.3390/ijms25105289

**Published:** 2024-05-13

**Authors:** Luisa Politano

**Affiliations:** Cardiomyology and Medical Genetics, University of Campania Luigi Vanvitelli, 80138 Naples, Italy; luisa.politano@unicampania.it

**Keywords:** heart involvement, DMD, BMD, LGMD, EDMD, MFMs, molecular gene defects heart transplantation, survival, life expectancy

## Abstract

Inherited muscular diseases (MDs) are genetic degenerative disorders typically caused by mutations in a single gene that affect striated muscle and result in progressive weakness and wasting in affected individuals. Cardiac muscle can also be involved with some variability that depends on the genetic basis of the MD (Muscular Dystrophy) phenotype. Heart involvement can manifest with two main clinical pictures: left ventricular systolic dysfunction with evolution towards dilated cardiomyopathy and refractory heart failure, or the presence of conduction system defects and serious life-threatening ventricular arrhythmias. The two pictures can coexist. In these cases, heart transplantation (HTx) is considered the most appropriate option in patients who are not responders to the optimized standard therapeutic protocols. However, cardiac transplant is still considered a relative contraindication in patients with inherited muscle disorders and end-stage cardiomyopathies. High operative risk related to muscle impairment and potential graft involvement secondary to the underlying myopathy have been the two main reasons implicated in the generalized reluctance to consider cardiac transplant as a viable option. We report an overview of cardiac involvement in MDs and its possible association with the underlying molecular defect, as well as a systematic review of HTx outcomes in patients with MD-related end-stage dilated cardiomyopathy, published so far in the literature.

## 1. Introduction

Inherited muscular diseases (MDs) are genetic degenerative disorders typically caused by mutations in a single gene that affect striated muscle and result in progressive weakness and wasting in affected individuals [1,2]. In addition, cardiac muscle can also be affected [3,4] with some variability that depends on the genetic basis of the MD phenotype. Most forms of cardiac involvement, such as Duchenne muscular dystrophy (DMD), myotonic dystrophy type 1 (DM1), and Emery–Dreifuss muscular dystrophy [EDMD], may be detected from childhood to the second decade of life; others can remain asymptomatic until later in life, including Becker muscular dystrophy (BMD), some forms of congenital myopathy, and myofibrillar myopathies (MFMs).

In some cases, cardiac involvement is more severe in forms with childhood onset [5]; however, the onset and progression of cardiac involvement are independent of the development of skeletal myopathy and may occur later [3,4]. Poor correlation exists between genotypes and phenotypes at the cardiac and skeletal muscle levels.

Cardiac and respiratory complications are the most frequent cause of death in patients with NMDs. However, while significant progress in the treatment of respiratory failure has led to increased survival and improved quality of life in these patients over the past several decades [6,7,8,9], cardiac involvement still remains a major cause of morbidity and mortality [10,11,12].

Clinically relevant cardiac involvement in MDs most commonly falls into one out of two major categories: cardiomyopathy and conduction defects with arrhythmias [3]. The severity and the onset of cardiac complications vary significantly across the different types of MDs [3,4]. Cardiomyopathy that evolves to congestive refractory heart failure is characteristic of dystrophinopathies (DMD, BMD, and X-linked DilatedCardioMyopathy (XL-DCM), females with dystrophinopathy), some forms of limb-girdle muscular dystrophies (LGMDR5, R6, R9, R13, and R14) [3], and congenital muscular dystrophies. Conduction system abnormalities, causing heart block, arrhythmias, and sudden cardiac death, are most commonly seen in DM1, laminopathies, and desminopathies, but they may similarly progress to dilated cardiomyopathy and intractable heart failure in the final stages. In these cases, heart transplantation (HTx) is considered the most appropriate option in patients who are not responders to the optimized standard therapeutic protocols [13]. However, in patients with inherited muscle disorders and end-stage cardiomyopathies, HTx is still considered a relative contraindication. High operative risk related to muscle impairment and respiratory failure, and the potential involvement of graft secondary to the underlying myopathy, have been the main reasons implicated in the generalized reluctance to consider cardiac transplant as a viable option.

We report an overview of cardiac involvement in MDs and its possible association with the underlying molecular defect, as well as a systematic review of HTx outcomes in patients with MD-related end-stage dilated cardiomyopathy, published so far in the literature.

### 1.1. Cardiac Involvement in Muscular Dystrophies

Table 1 lists muscular dystrophies analyzed in this review.

#### 1.1.1. Dystrophinopathies

Dystrophinopathies belong to the large spectrum of muscular dystrophies and present a variable involvement of skeletal and cardiac muscles; they are due to mutations in the *DMD* gene, which encodes the dystrophin protein [14]. Depending on the type of mutation, patients can develop the severe form of muscular dystrophy (Duchenne, DMD), the milder phenotype (Becker muscular dystrophy, BMD) [15,16], or a rapidly evolving dilated cardiomyopathy (XL-DCM), in which the heart is predominantly or exclusively affected [17]. 

Genetic testing is highly sensitive and specific. In about 65–75% of cases, the diseases are due to large deletions; in the remaining 25%, they are due to duplications or point-mutations [18,19]. Suspicion of a dystrophinopathy should arise in any patient with very high serum creatine kinase (CK) levels associated with signs of muscle weakness. 

The picture of severe dilated cardiomyopathy with intractable heart failure is typical of BMD [20,21,22,23] XL-DCM [17] and DMD/BMD carriers [24,25,26] and frequently observed in patients with DMD over the age of 18 [27,28]. Conduction system defects and supraventricular or ventricular arrhythmias may also be present, especially in the end stages of the disease. 

The management of dystrophinopathies is primarily based on supportive care. Glucocorticosteroids (GCS) are accepted as the standard of care in DMD, as they have been shown to have changed the natural history of DMD by prolonging ambulation, improving cardiac health, and increasing long-term survival [29,30,31]. Pharmacological treatment for cardiac manifestations includes the use of angiotensin-converting enzyme inhibitors (ACEi) or angiotensin receptor blockers (ARBs), beta-blockers, diuretics, and antiarrhythmic drugs as standard treatments of dilated cardiomyopathy and arrhythmias [32,33,34,35,36]. An exciting and ongoing area of investigation is the gene therapy to re-express dystrophin and slow the rate of muscle degeneration [37,38,39]. 

#### 1.1.2. Emery–Dreifuss Muscular Dystrophies

Emery–Dreifuss muscular dystrophy (EDMD) is a rare muscle disorder, characterized by association with frequent life-threatening cardiac complications. The initial description of EDMD by Emery and Dreifuss in 1966 [40] classically presents with muscle weakness, early contractures, atrial paralysis, and fibrillation. According to the pattern of inheritance, two main forms of EDMD can be distinguished, an X-linked form (XL-EDMD, EDMD1) caused by mutation in emerin gene (*EDM*) [41,42], and an autosomal dominant form (AD-EDMD, EDMD2) caused by mutations in *LMNA A/C* gene [43]. In both forms, cardiac conduction abnormalities with different degrees of atrio-ventricular block, atrial or ventricular arrhythmias that often require ICD implant, and severe cardiomyopathy are frequently observed [44,45,46,47,48,49,50,51,52,53,54]. EDMD2 may present with a wide range of cardio-muscular phenotypes, from the classical form described in 1966, to a limb-girdle muscular dystrophy-like form, or to predominant/exclusive cardiac involvement evolving to dilated cardiomyopathy and refractory heart failure [44,45,46,47,48,49,50,51,52,53,54]. Patients with EDMD are at high risk for cardiac sudden death [55,56,57,58,59].

#### 1.1.3. Limb-Girdle Muscular Dystrophies

Limb-girdle muscular dystrophies (LGMDs) are a highly heterogeneous group of genetic muscle disorders, which first affect the voluntary muscles of the hip and shoulder areas. Signs and symptoms may begin at any age and usually worsen by time. Childhood forms resemble the DMD phenotype, with loss of ambulation in the first–second decade of life, while adult forms present a more benign evolution. LGMDs are autosomal disorders with both dominant and recessive inheritance and their prevalence is not the same in different areas [60,61]. The first description as a disease clinically distinguished from the forms of BMD and facioscapulohumeral dystrophy (FSHD) dates back to 1954 by Walton and Natrass [62] in their classification of muscle diseases. However, for the following 40 years, the group of LGMDs was considered as a “cauldron” into which all those cases of muscular dystrophy that did not have a well-defined clinical picture were included. Between the late 1980s and 1990s, thanks to the papers published by Rideau in France [63,64,65] and the discovery of new genes responsible for muscle diseases, LGMD was recognized as a separate clinical entity. 

Ten autosomal dominant and 25 autosomal recessive limb-girdle muscular dystrophies have been identified (Table 2) in an initial classification of the different types of LGMD, based on the chronological discovery of the related genes. The classification divided the different forms of LGMDs according to the pattern of inheritance into autosomal dominant (LGMD1) and autosomal recessive (LGMD2) forms, followed by a consecutive letter of the alphabet. Thus, for example, recessive LGMD caused by mutations in the calpain 3 gene, the first identified, was called LGMD2A; that caused by mutations in the dysferlin gene (the second identified), LGMD2B; and so on [66,67].

The acceleration in the discovery of new genes, thanks to next generation sequencing (NGS) [68] and the diagnostic refinement of phenotypes, has made the number of letters of the alphabet insufficient. Therefore, it was necessary to introduce new classifications that are based on deficient protein (calpainopathies, dysferlinopathies, etc.), or that follow the formula “LGMD, inheritance (recessive or dominant), order of discovery (number), affected protein” (LGMDR1, LGMDD1, etc.) [69]. Table 2 shows the new and old nomenclature of LGMDs according to the proposed classifications. The new classification identifies five dominant forms of LGMDs, compared to the previous 10, and 25 recessive forms. Some dominant forms, such as myotilin (1A) and desmin (1E) defects, are now included in the group of myofibrillar myopathies.

Patients suffering from LGMDs also present a multisystem disease that requires a multidisciplinary approach, in which cardiac and respiratory involvement dominate, with variable prevalence within the different types. Cardiac involvement can be present both as a degenerative process characterized by fibrosis and fatty replacement of the myocardium with evolution towards dilated cardiomyopathy and refractory heart failure, and as conduction tissue defects, characterized by atrioventricular blocks or severe tachyarrhythmias. The two conditions may coexist.

The LGMD forms most frequently associated with cardiac involvement are LGMD1B (replaced by EDMD2) [44,45,46,47,48,49,50,51,52,53,54] and LGMDR9 [70,71,72,73,74,75,76]. LGMDR9 manifests with a wide clinical variability and evidence suggests that this, in part, reflects the genotype: subjects homozygous for c.826C>A gene variant express a milder phenotype compared to compound heterozygous subjects, but can develop severe cardiomyopathy [71,74]. Cardiomyopathy has also been observed in patients affected by sarcoglycanopathies (LGMDR3-R6) [77,78,79,80,81,82,83,84,85,86,87] and LGMDR13 [88,89,90] in percentages ranging from 12.9 to 50; the tendency to evolution towards dilated cardiomyopathy and heart failure is more frequently observed in patients with EDMD2, LGMDR6, R9, R11, and R13 [44,45,46,47,48,49,50,51,52,53,54,55,56,57,58,59,60,61,62,63,64,65,66,67,68,69,70,71,72,73,74,75,76,77,78,79,80,81,82,83,84,85,86,87,88,89,90]. Isolated cases of cardiac involvement have also been described in patients with LGMDR7 (defects in telethonin), R10 (defects in titin) [91], and R23 (defects in *LAMA 2* gene) [92]. 

#### 1.1.4. Myotonic Dystrophy Type 1

Myotonic dystrophy type 1 (DM1) or Steinert’s disease, first described in 1909 [93] is the most common muscular dystrophy in adult life, with an estimated prevalence of 1:8000 [94,95]. DM1 belongs to the group of trinucleotide repeat disorders caused by CTG triplet expansion in the 3′ non-coding region of the *DMPK* gene, which encodes the DM protein kinase [96].

Cardiac involvement, represented mainly by arrhythmias and conduction tissue disorders, significantly contributes to the morbidity and mortality of these patients, who have an increased risk of sudden cardiac death due to complete atrioventricular block or tachy-arrhythmias [23,97,98,99,100,101]. Cardiac pacing is often required [98,99,101]. However, ventricular dysfunction associated with conduction disorders has also been reported [102]. The prevalence of left ventricular systolic dysfunction (LVSD), defined as left ventricular ejection fraction (LVEF) < 55%, is estimated at 13.8%, which is 4.5 times higher than in the general population [102]. Patients with DM1 and LVSD were predominantly male, older, and had a longer atrioventricular and intraventricular conduction duration at baseline; they also had a higher prevalence of atrial arrhythmias, and implanted more frequently. Furthermore, symptomatic heart failure was frequent despite the limited levels of physical activity [102]. Severe ventricular systolic dysfunction (VSD) is rare and usually occurs late in the course of the disease [99,101,102].

#### 1.1.5. Myofibrillar Myopathies

Myofibrillar myopathies (MFMs) represent a group of muscular dystrophies with a similar morphological phenotype [103,104,105]. The diagnosis is established on muscle biopsy. The MFMs are characterized by a distinct pathological pattern of myofibrillar dissolution associated with disintegration of the Z-disk, accumulation of myofibrillar degradation products, and ectopic expression of multiple proteins, such as desmin, αB-crystallin, dystrophin, and, sometimes, congophilic material [103,104,105]. So far, pathogenic mutations associated with MFM phenotype, including atypical MFM-like cases, have been identified in 17 genes: *DES, CRYAB, MYOT, ZASP, FLNC, BAG3, FHL1, TTN, DNAJB6, PLEC, LMNA, ACTA1, HSPB8, KY, PYROXD1,* and *SQSTM + TIA1* (digenic). Most of these genes are also associated with other forms of muscle diseases. In addition, in many MFM patients, numerous genomic variants in muscle-related genes have been identified [106,107]. The clinical features of MFM are highly variable. These include progressive muscle weakness that often involves or begins in distal muscles, but girdle or scapula-fibular distributions may also occur [103,104,105]. Desminopathies represent a discrete percentage of patients with myofibrillar myopathies [108,109]. Cardiac involvement is very common in autosomal dominant desminopathies and can sometimes be the initial or only symptom of the disease [110,111,112]. Arrhythmias and arrhythmogenic cardiomyopathy are frequent associated features [113,114]. However, autosomal recessive desminopathies with associated cardiomyopathy were also reported [115]. No curative treatment for MFMs is currently available, so careful follow-up, especially of cardiac and respiratory function, is important [23,35].

#### 1.1.6. Gender Differences

Gender differences regarding muscle aspects are reported in numerous limb-girdle dystrophies. For example, in those due to defects in Calpain 3 [116], anoctamine 5 [117,118], and teletonin [119], males are more severely affected than females, while in those caused by defects in dysferlin [120] or *FKRP* [121], females are more affected than males. On the contrary, cardiological aspects have rarely been investigated and little information is available in the literature, probably because the most frequent and severe forms of muscular dystrophy such as DMD, BMD, and EDMD1 usually affect males. Only two papers deal with this topic in the literature. The first [121] reports that male patients with R9 LGMD, homozygous for the c.826C>A mutation, have a positive correlation with the onset of cardiomyopathy, but not with age or stage of the disease. The second [122], which concerns patients with sarcoglycanopathies, reports no correlation with sex but only with the duration of the disease.

### 1.2. Association of Cardiac Disease to Molecular Defects

The association of cardiac involvement in patients with muscular dystrophies has been known for a long time; however, the clinical and genetic heterogeneity of these conditions had hindered, until recently, the ability to recognize individual disease complications. The improved understanding of the molecular defects responsible for these forms over the years has facilitated the recognition of the associated cardiac complications. In this light, a possible correlation between myocardial involvement and molecular defects was investigated for those muscle diseases in which cardiomyopathy or conduction defects is most frequently observed, with the intent to diagnose patients presenting these variants early and accurately, and to implement subtype-specific anticipatory treatments. 

Figure 1 shows a simplified schematic representation of the proteins associated with sarcolemma in skeletal muscle fibers. Most of them are located at the sarcolemmal level of the myofibers. This is the case with sarcoglycan complex, integrins, dysferlin, caveolin, sarcospan, and beta dystroglycan. Other proteins are located on the extracellular matrix (collagen VI, laminin 2, alpha dystroglycan, and fukutin), or act as a bridge between the two structures such as dystrophin; still others are part of the cytoskeleton (actin and myosin) or myofibrils (teletonin, desmin, etc.) or are located in the internal membrane of the nucleus, or inside it (emerin and lamina A/C).

#### 1.2.1. Dystrophinopathies

Dystrophinopathies are caused by mutations in *DMD* gene, the largest known human gene, covering 2.4 megabases (0.08% of the human genome) at locus Xp21 [123]. The 79-exon muscle transcript codes for a protein of 3685 amino acid residues [14]. Mutations causing diseases include deletions, duplications, and point mutations. The phenotype, BMD or DMD, depends on whether these mutations allow the preservation of the DNA reading frame or not [124]. Dystrophin acts as a bridge between sarcolemma and actin cytoskeleton. Though representing only 4% of all muscle proteins, it plays a fundamental role in protecting the membrane from the stress induced by muscle contraction, and in regulating the function of dozens of muscle proteins on the sarcolemma [14].

Severe dilated cardiomyopathy with refractory heart failure accounts for the highest number of deaths in patients with BMD and DMD [3,10,11,20,27]. In a group of 19 patients with BMD, we noted that cardiomyopathy was predominantly associated with deletions including exons 45–48, and never associated with deletions not exceeding the exon 47 [125]. This observation allowed us to suggest a possible correlation between the severity of cardiomyopathy and particular *DMD* gene deletions [125]. This type of correlation was subsequently confirmed by us in a study involving a larger group of 284 patients with dystrophinopathies [126], and by Melacini et al. [127], who found that deletions involving exon 49 were invariably associated with cardiomyopathy, whereas deletion of exon 48 was associated with heart disease in the highest number of patients with BMD investigated. 

In 2009, Kaspar et al. [128] analyzed 78 patients with BMD and XL-DCM, with common deletions predicted to alter the dystrophin protein, and correlated these mutations to the age of onset of cardiomyopathy. They found that deletions affecting the amino-terminal domain (deletions of exons 2 to 9 of the *DMD* gene) are associated with early-onset DCM (mid-20s), while deletions removing part of the rod domain and hinge 3 (deletions of exons 45 to 49) have a later onset DCM (mid-40s). By combining genetic and protein information, their analysis revealed a strong correlation between specific protein structural modifications and age of onset of dilated cardiomyopathy. In particular, they found that cardiac dystrophin may be particularly sensitive to structural disruptions of the exon 45–49 region compared to skeletal muscle dystrophin. Their conclusions were in agreement with the studies in dystrophin-null mdx mice expressing a mini-dystrophin construct that lacked the exon 45 to 49 region but had an intact hinge 3 domain. In these mice, only a partial restoration of cardiac function was achieved in spite of a complete rescue of the skeletal muscle pathology [129]. Restrepo-Cordoba et al. [130], in their series of 112 patients with DCM associated to *DMD* gene mutations, with and without muscle impairment, have recently shown no correlation to the type of mutation and no differences between type of mutations (truncating and non-truncating variants) and the mean age at DCM diagnosis [130]. However, it should be noted that the two papers cannot be compared because they consider two different patient populations and have different objectives. Kaspar et al. [128] studied a population of patients with the Becker phenotype and analyzed in detail the association type of deletion-onset of cardiomyopathy. In contrast, Restrepo-Cordoba et al. [130] analyzed a population of patients with the cardiac phenotype (dilated cardiomyopathy) associated with mutations in the *DMD* gene. Moreover, among the 79/112 patients who also had skeletal myopathy, only six shared the deletion 45–49 considered crucial by Kaspar et al. [128]. Furthermore, the Restrepo-Cordoba’s study aimed to describe the prognosis of dystrophin-associated DCM in patients without skeletal myopathy, underlining how these patients should be offered lifelong surveillance to diagnose and manage cardiac complications.

#### 1.2.2. Emery–Dreifuss Muscular Dystrophies

Mutations in the nuclear envelope proteins emerin and lamin A cause a number of diseases including premature aging syndromes, muscular dystrophy, and cardiomyopathy [131]. Emerin and lamin A are implicated in regulating muscle- and heart-specific gene expression and nuclear architecture. Additionally, emerin and lamin A are also required to maintain nuclear envelope integrity [131].

Lamins A and C are components of the nuclear envelope but are located in the lamina, a multimeric structure associated with the nucleoplasmic surface of the inner nuclear membrane (Figure 1). These highly conserved proteins are transcribed from a single gene, *LMNA*, that is encoded on chromosome 1q21.2–q21.3.29. Lamins are structurally homologous with other intermediate filaments and consist of a central-rod domain flanked by globular amino and carboxyl domains. *LMNA*-related cardiomyopathy explains 5–10% of familial DCM and 2–5% of sporadic DCM [132,133]. The clinical course of DCM is considered to be worse among patients harboring *LMNA* mutations than among those without LMNA mutations [134]. Several groups reported that sudden cardiac death, atrioventricular block, or fatal ventricular tachycardia is observed in *LMNA*-related cardiomyopathy even though the left ventricular systolic function is still preserved [135,136].

More than 160 different mutations in the *LMNA* gene have been identified as a cause of cardiomyopathy [137,138,139]. They include missense, nonsense, in-frame and out-of-frame insertions/deletions, splice site mutations and, rarely, large exonic deletions. Among them, missense mutations are by far the most frequent type of mutations observed in laminopathies [138,139,140], and those occurring in the rod domain cause dilated cardiomyopathy and conduction-system disease [48]. Lamin mutations leading to DCM are rarely found in the tail domain, where linked mutations to EDMD, familial partial lipodystrophy, and Hutchinson–Gilford progeria syndrome are instead observed [141]. However, no hot spots can be identified for DCM or muscular dystrophies and, due to the high phenotypic variability among subjects sharing the same lamin A/C gene mutation, it is not possible to predict the extent and severity of cardiac involvement based on the characteristic of the mutation [142]. In an Italian retrospective study of a cohort of 78 patients with laminopathy [140], frameshift mutations were more frequently detected in patients with LGMD phenotype, and in those with heart involvement. 

Holt et al. [143], using transfection of lamin-A/C-deficient fibroblasts, studied the effects of nine pathogenic mutations on the ability of lamin A to assemble normally and to localize emerin normally at the nuclear rim. Five mutations in the rod domain (L85R, N195K, E358K, M371K, and R386K) affected the assembly of the lamina. With the exception of mutant L85R, all rod domain mutants induced the formation of large nucleoplasmic foci in about 10% of all nuclei [143]. The presence of emerin in these foci suggests that the interaction of lamin A with emerin is not directly affected by the rod domain mutations. Three mutations in the tail region, R453W, W520S, and R527P, might directly affect emerin binding by disrupting the structure of the putative emerin-binding site, because mutant lamin A localized normally to the nuclear rim but its ability to trap emerin was impaired [143]. They hypothesized that mutations in the tail domain of lamin A/C work by direct impairment of emerin interaction, whereas mutations in the rod region cause defective lamina assembly that might or might not impair emerin capture at the nuclear rim [143].

Furthermore, loss of function (LoF) variants in *LMNA* gene cause prevalently dilated cardiomyopathy with associated conduction defects [144]. 

#### 1.2.3. Limb-Girdle Muscular Dystrophies

The high clinical and genetic heterogeneity of this group of muscle disorders hinders the possibility of genotype–phenotype correlations at the heart level. In a large European cohort of 396 patients with sarcoglycanopathies [85,145,146], no correlation was found between specific gene mutation and the onset of cardiomyopathy. However, novel findings seem to indicate the involvement of exon 3 in all compound-heterozygous patients with alfa-sarcoglycanopathy who suffered from cardiomyopathy [145]. 

The mutation c.826C>A (p. L276I), a common founder *FKRP* mutation, is present in at least one allele in most patients with LGMDR9 [147]. However, the relationship between the development of cardiomyopathy and this mutation genotype is controversial. One study [71] suggested that individuals homozygous for the common c.826C>A FKRP mutation are at greater risk of developing cardiomyopathy than those with compound heterozygous mutations, while other studies [70,148] proposed the reverse relationship. The discordant results between the studies could depend on the number of patients investigated, as well as on the prevalence of the homozygosity condition in some countries or on having included other types of mutations in the heterozygous group.

#### 1.2.4. Myotonic Dystrophy Type 1

Myotonic dystrophy type 1 (DM1) is the most common adult-onset muscular dystrophy, presenting as a multisystemic disorder with extremely variable clinical manifestation, from asymptomatic adults to severely affected neonates. It is an autosomal dominant hereditary disease associated with an unstable expansion of CTG repeats in the 3′-UTR of the *DMPK* gene, with the number of repeats ranging from 50 to several thousand. There is agreement that the number of CTG repeats positively correlates with both the age-at-onset and overall severity of the disease [149]. 

Numerous studies have investigated the existence of a correlation between the size of the repeats and the age-at-onset and severity of the cardiac phenotype. Melacini et al. [150], by studying a group of 42 adults with DM1, reported that abnormal ventricular late potentials were directly correlated with CTG expansion (*p* = 0.05) and that the incidence of ventricular couplets or triplets showed a positive correlation with size of CTG expansion. In 2001, Finsterer et al. found that in patients 21–50 years of age, cardiac involvement increased with increasing CTG-repeat size [151].

A recent French study [152], which included 985 patients from the Cochin Hospital Registry on Myotonic Dystrophy type 1, reported that CTG expansion size significantly correlated with heart rate. The prevalence of conduction system disease and left bundle branch block on the ECG was significantly higher, and PR and QRS intervals were longer, in patients with larger mutations [152]. By contrast, no difference was observed for left ventricle (LV) ejection fraction, or for the prevalence of LV systolic dysfunction, supraventricular arrhythmias, atrial fibrillation, or premature ventricular contractions [152]. A positive correlation between repeats length and diastolic heart dysfunction was also found by Park et al. [153]. 

However, the association between mutation size (CTG expansion) and severity of cardiac involvement remains controversial, as some authors believe that the extent of expansion studied at the blood level may not reflect the size of expansion in different organs and tissues, leading to erroneous correlations.

#### 1.2.5. Desminopathies

Mutations in the *DES* gene, which encodes desmin, can cause myopathies in general, and familial and sporadic cardiomyopathies in particular [154]. The majority of *DES* mutations are heterozygously inherited, indicating a dominant negative genetic mechanism or putative haploinsufficiency [154,155,156]. Most patients with desminopathy have dominant de novo in-frame mutations in *DES* gene [108]. 

However, some rare cases with compound heterozygous or homozygous *DES* truncating mutations were also described indicating that, in specific cases, the inheritance can be also recessive [109]. In a recent review of all cases of desminopathy with autosomal recessive inheritance, Onore et al. [115] confirmed that these are more severe than dominant ones, as patients with loss of function (nonsense or out-of-frame) bi-allelic variants develop clinical manifestations much earlier than those with dominant forms, and have a higher risk of premature sudden cardiac death. However, as severe cardiorespiratory involvement and premature death can be similarly caused by compound heterozygous or homozygous non-truncating variants (e.g., missense variants or small indels that do not cause frameshift), a genotype–phenotype correlation is not currently possible [157,158]. A meta-analysis focused on phenotype–genotype correlation revealed that the isolated cardiac phenotype was found more frequently in patients with a mutation in the head or the tail domain [159]. 

Two founder mutations (p.S13F and p.N342D) were identified in Netherlands [159,160]. Mutation p.S13F (c.38 C>T) is a missense mutation in the head domain, while mutation p. N342D (c.1024A>G) is a missense mutation in the 2B domain. The cardiac phenotype was the presenting symptom in about 80% of patients carrying p.S13F mutation [159] and included sudden cardiac death or progressive heart failure. Cardiac involvement is fully penetrant and severe, characterized by cardiac conduction disease and cardiomyopathy, often with right ventricular involvement [159,160]. About 36% of known and obligate carriers died, underwent transplantation, or experienced appropriate implantable cardioverter defibrillator (ICD) interventions, at a mean age of 48.4 years [159,160]. Mutation p. N342D was associated with muscle symptoms. Heart involvement was less frequent and specific [160]. 

Two further novel desmin mutations—p.N116S, which is located in segment 1A of the desmin rod domain [161], and p.Glu401Asp, which is located in segment Coil 2B [162]—have recently been reported associated with inherited left ventricular arrhythmogenic cardiomyopathy/dysplasia, with a high incidence of adverse clinical events in the absence of skeletal myopathy or conduction system disorders. The first mutation, p.N116S, was identified in a patient with arrhythmogenic right ventricular cardiomyopathy and terminal heart failure [161]; the second mutation, pGlu401Asp, was found in a large Spanish family in which 30 individuals presented with an arrhythmogenic phenotype with high risk of sudden cardiac death and progressive heart failure [162]. The pathogenic mechanism was related to a probable alteration in desmin dimer and oligomer assembly and its connection with membrane proteins within the intercalated disc [161,162].

Fisher et al. [163] in a large German family sharing the p. Glu410Lys mutation have recently shown, that cardiac desminopathies could involve both ventricles, and that the arrhythmogenic cardiomyopathy has a phenotypic spectrum, even in the same family, which ranges from ARVC and DCM to isolated AV-block. The patient with the p.Ile402Thr mutation instead had DCM in combination with skeletal myopathy. The overlap with the dilated cardiomyopathy phenotype makes adequate monitoring and follow-up necessary in these patients, due to the risk of potentially lethal arrhythmias, regardless of left ventricular dysfunction [164,165]. Finally, Protonotarious et al. have recently reported that the novel desmin variant p.Leu115Ile is associated with a unique form of biventricular arrhythmogenic cardiomyopathy [166].

## 2. Methods

### Search and Selection

An extensive literature search using a combination of three major medical literature databases: Pubmed, Scopus, and Embase was performed. The search considered all full-text articles that were available up to July 2023. No restriction was applied to the study type or design. The articles were selected based on the following inclusion criteria: −inclusion of patients with muscular dystrophies in the cohort;−systematic reviews, prospective and retrospective cohort studies including case series to capture all published material;−defined outcomes, such as dilated heart transplantation and survival.

The main topics for the search were heart transplantation, survival, life expectancy, muscular dystrophies, Duchenne muscular dystrophy (DMD), Becker muscular dystrophy (BMD), Emery–Dreifuss muscular dystrophy (EDMD), limb-girdle muscular dystrophy (LGMD), myotonic dystrophy type 1 (DM1), myofibrillar myopathies (MFMs), and desminopathies (DES).

The literature search was updated in October 2023 and yielded 260 results, 175 in PubMed, 85 in Scopus, and 0 in Embase. After removal of duplicates, 245 articles were identified, and, after exclusion of articles for inappropriate topic, 172 articles. After screening of title and abstract, 73 studies were found eligible for full article review. Twenty-one records were excluded because not compliant with abstract. Fifty-two studies were included in the analysis (Figure 2).

## 3. Results and Discussion

Systematic review of heart transplantation (HTx) in muscular dystrophies.

### 3.1. Dystrophinopathies

#### 3.1.1. Duchenne Muscular Dystrophy

Of the 11 heart transplants reported in patients with DMD, nine are part of three larger series [167,168,169], and two are case reports [170,171]. Rees et al. [167], in 1993, first reported heart transplantation in six patients with muscular dystrophies, three of them having DMD. The mean age at transplant in all patients was 22.3 (range 12–31 years) and the follow-up (FU) 2.8 (range 10 months–5.4 years). All patients received triple-drug immunosuppression, consisting of azathioprine, cyclosporine, and steroids, and had an uneventful postoperative course. 

The second report was published 17 years later by Wu et al. [168] and involved 29 patients with MDs, 15 of whom had BMD, 1 EDMD, 3 LGMD, 4 DM1, or 3 DMD. The mean age at transplant of the entire group was 37 years, and the average FU 5.4. Of the three transplant recipients with DMD, each had one episode of rejection within the first year of transplantation, while one patient had an episode of bacterial infection, followed by cytomegalovirus infection, within two months of transplant, and the third developed allograft vasculopathy 10 years after transplantation. All were alive at the end of the follow-up period (10 years). Wells et al. [169] compared the results of heart transplantation in a cohort of 81 patients with neuromuscular diseases (3 DMD, 42 BMD, 11 EDMD, 4 LGMD, 2 DM1, 1 CMT, and 18 with unspecified type) with a cohort of all cardiomyopathies. The three patients with DMD were transplanted at a mean age of 15.7 (range 14–18 years) and had an average follow-up of 3.6 years (range 1.9 -10). One of them died of bacterial septicemia 3.6 years after surgery.

The two single cases were reported by Cripe et al. in 2011 [170] in a 14-year-old boy, with a FU of 4 years and by Piperata et al. [171] who described the case of an 18-year-old DMD patient who underwent successful heart transplantation after 47 months of HeartWare Left Ventricle Assist Device (LVAD) assistance. The three months’ follow-up was uneventful.

#### 3.1.2. Becker Muscular Dystrophy and XL-DCM

End-stage dilated cardiomyopathy with refractory heart failure is the main cause of death in patients with BMD or XL-DCM who often have preserved muscle function and almost autonomous walking until the sixth–seventh decade. For this reason, more than 1/3 of all patients with DMs who undergo cardiac transplantation are patients with BMD. One-hundred-sixteen cases of heart transplantation are overall reported in patients with BMD. Before the 1990s, only two cases were described, one by Casazza et al. in 1988 [172], and the other by D’Onofrio et al. in 1989 [173]. The age of implantation was 23 years in the first patient and 17 years in the second, with a similar follow-up of 2 years. In the period 1990–1999, eight cases of cardiac transplantation in patients with BMD were published [167,174,175,176,177,178]. Data regarding age at transplantation and duration of FU are available for all patients but three [174]. The average age of HTx was 23.6 (range 15–32 years) and the average FU was 3.2 years (range 0.6–6). Most reports are single cases, with the exception of Rees’ paper [167], which found one patient with BMD out of a total six patients with MDs associated with end-stage cardiomyopathy. All patients received triple-drug immunosuppression consisting of azathioprine, cyclosporine, and steroids, and had an uneventful postoperative course. Postoperative intubation time was no longer in these patients compared to other recipients. All patients were physically rehabilitated. One patient died suddenly 27 months after transplantation. Annual re-catheterization studies showed normal left ventricular ejection fraction. No signs of coronary artery disease were observed, nor any progression of pre-existing muscular dystrophy, at the time of the study.

In the following decade (2001–2010), there were 25 cases of cardiac transplantation in patients with BMD [179,180,181,182,183,184,185,186], seven of them case reports [179,180,182,183,184,185,186] and two reporting on 15 [168] and four [187] patients, respectively. The age of transplant in single patients was 26.2 years (range 14–38) and FU of about 2 years for patients for whom these data are available. Age at transplantation and FU are not available for the 15 patients reported by Wu et al. [168], who indicate, for the entire group of patients with muscle diseases, an average age at transplant of 37 years and a FU greater than 10 years. The three patients reported by Ruiz-Cano et al. [181] had an average age at transplant of 39.5 (range 24–55) and an average FU duration of 4.8 years (range 1.1–10.8).

In the period 2011–2023, 81 cases [130,188,189,190,191,192] of HTx in patients with BMD were reported in the literature. One concerns a single case [188] while the other six papers reported 15, 42, 7, 4, 6, and 6 cases, respectively. Restrepo-Cordoba et al. [130], in a series of 112 patients with mutation in the *DMD* gene, found that DMD-associated dilated cardiomyopathy without severe skeletal myopathy was characterized by a high risk of major adverse cardiac events, including progression to end-stage heart failure and ventricular arrhythmias. Among them, 15 patients needed a heart transplant at a mean age of 35 (range 19–51) and were followed for a mean period of 8 years. Wells et al. [169] confirmed these data by comparing the results of heart transplantation in a cohort of 81 patients with MDs, 42 of them with BMD, with a cohort of all cardiomyopathies. An overall mean age of 22 years (range 15–33) at heart transplant has been reported for the entire group, but no mention is made about the duration of FU. The authors concluded that patients with MDs undergoing HTx have similar long-term post-transplant survival. Furthermore, no significant differences they observed between BMD (the larger group) and other forms of muscular dystrophy, suggesting that HTx may be an effective treatment for a selected group of patients with muscular dystrophy and end-stage heart failure [169].

Steger et al. [188] reviewed seven cases of BMD who were transplanted for end-stage cardiomyopathy at a mean age of 38.5 (range 16–56 years) and a mean duration of FU of 5.7 (range 1.4–11.6 years). Papa et al., in 2017 [187], published a review on heart transplantation in patients with end-stage-dystrophin-related cardiomyopathy and reported the personal experience of four patients with BMD/XL-DCM, who were transplanted at a mean age of 30.2 years (range 27–34) and were post-operatively followed for 12.0 years (range 10–14.5). They reported that long-term clinical outcomes of heart transplantation in appropriately selected patients are similar to those of a matched cohort of patients undergoing heart transplantation for idiopathic dilated cardiomyopathy. In particular, they recommended intensive neuro-muscle observation to adequately adjust the dosage of immune suppressants and avoid the onset of secondary myopathy, and careful monitoring of the onset of rhabdomyolysis due to the toxic effect of ciclosporin, especially when the latter is used in combination with lipid-lowering agents, such as statins and gemfibrozil, as indicated by Ketelsen et al. [189]. The paper provides evidence that heart transplant is a safe and effective treatment for selected patients with end-stage-dystrophin-related cardiomyopathy [187].

Ascencio-Lemus et al. in 2019 [191] reported the successful results of HTx in a group of six patients with BMD. The average age at the transplant was 38.9 years (range 24–48 years) and the mean duration of FU was 8.4 years (range 0.10–19.1 years).

Recently, Visrodia et al. [192] reported the results of HTx in 20 patients with MD, six with BMD, four with EDMD, four with LGMD, two with DM1, and four with other muscle pathologies. They state that the survival rate at 5 years is comparable to that of a cohort of non-MD HTx recipients (95% versus 100%) and that only two patients (10%) experienced worse mobility after transplantation.

#### 3.1.3. DMD/BMD Carriers

Only three cases of heart transplantation in DMD carriers have so far been published. The first case was reported in 1998 by Melacini et al. [193] in a 41-year-old female with positive history of DMD, who was followed for 3.6 years. The second case was reported by Davis et al. in 2001 [194] and concerned a 25-year-old carrier of Duchenne’s muscular dystrophy who developed severe cardiac failure and required mechanical circulatory support and transplantation. The third case was recently published by Cullom et al. in 2022 [195] and concerned a 60-year DMD symptomatic carrier with end-stage cardiomyopathy with refractory heart failure, who was in a good health state 2 years after HTx. 

### 3.2. Emery–Dreifuss Muscular Dystrophies

Patients affected by EDMD2/LGMD1B more frequently undergo cardiac transplantation because they present potentially lethal arrhythmias requiring PM/ICD implantation, and evolution towards ventricular dilation and cardiac failure, often without an overt skeletal muscle involvement. This is the reason why a greater number of publications, both as case reports and as case series, are present in the literature. 

Before the 1990s, no case of HTx in patients with EDMD was reported. Between the 1990s and 2000, only three case reports of HTx were published. The first case was reported in 1990 by Merchut et al. [196], who described a young woman with humero-fibular muscular dystrophy and contractures (EDMD2) receiving heart transplant for severe dilated cardiomyopathy. Cardiac histopathology showed myocyte hypertrophy, interstitial fibrosis, and nuclear hyper-chromaticity without mitochondrial abnormalities. Her myopathy and heart disease were not clinically evident in her family, although three of her relatives had inexplicably shortened Achilles tendons without weakness. The authors underlined how dilated cardiomyopathy can also occur in the autosomal dominant form of EDMD (EDMD2), and evolve into severe congestive heart failure. In the same year, Anthuber et al. [175] reported a second successful heart transplantation in a 47-old EDMD patient with a post-transplant FU of 3.6 years. The third case was included among the cases reported by Rees [167] and concerned a patient transplanted at 47 years and still alive after 7 years of FU. 

In the period 2000–2010, 22 cases of Htx in patients with EDMD were reported [197,198,199,200,201,202,203]. Kichuk-Chrisant et al. in 2004 [197], reported a successful heart transplant in two identical twin brothers with AD-EDMD, complicated by ventricular arrhythmias and end-stage cardiomyopathy. They suggested that early recognition of progressive heart disease and subsequent heart transplantation could be lifesaving in patients with EDMD. In 2005, Ben Yahou et al. [198] reported that 2/14 patients (1.4%) from eight families with laminopathy needed a heart transplant. Kärkkäinen et al. [199] found that six Finnish patients who survived a heart transplant for dilated cardiomyopathy between 1984 and 1998 had a mutation in the *LMNA* gene. They recommended screening for this gene at least among patients with conduction system disorders. Cuneo et al. [200] reported the case of a 21-year-old young man, diagnosed with AD-EDMD 3 years earlier, who underwent an orthotopic heart transplant for dilated cardiomyopathy and congestive heart failure. Dell’Amore et al. [201] reported on two patients with EDMD who received heart transplantation at 43 and 45 years, respectively. The average duration of FU was 4.5 years (range 3.4–5.6). Ambrosi et al. [202] described seven heart transplant recipients in a single family with mutation in *LMNA* gene. These patients showed no early or late post-operative higher complications than other heart transplant recipients at a mean follow-up of eight years (range 1–17 years). Furthermore, no case of rhabdomyolysis or worsening of musculoskeletal symptoms was observed. A further case of HTx in a patient with EDMD, was reported by Wu et al. in 2010 [168]; however, no specific information about age and FU of this patient is available. In the same year, Volpi et al. [203] reported a successful heart transplantation in a 58-old-female with a rare *LMNA* gene duplication.

Between 2011 and 2023, 73 cases of HTx in patients with EDMD were reported in the literature [48,50,58,169,192,204,205,206]. Apart from three single cases [204,205,206] concerning patients with EDMD who received HTx at the age of 38, 33, and 32, respectively, the other four papers concern large patient series. Hasselberg et al. [48] reported heart transplantation in 15 of 79 (19%) *LMNA* patients during 7.8 ± 6.3 years of follow-up. Ditaranto et al. [50] investigated differences in cardiac phenotype and natural history in 40 *LMNA*-mutated patients with and without neuromuscular onset. They found 26 cases with cardiac clinical presentation and 14 with neuromuscular presentation. The latter had early symptoms in life, developed atrial fibrillation/flutter (AF) and required pacemaker implantation at a younger age. However, they developed cardiomyopathy less frequently and had a lower rate of sustained ventricular tachy-arrhythmias compared to patients with cardiac onset. Rhythm disturbances usually occurred before evidence of cardiomyopathy, but despite these differences, the need for a heart transplant and the mean age at surgery were similar in the two groups. Peretto et al. [58], in a series of 164 Italian patients with mutations in *LMNA* gene, followed for an average period of 10 years (range 7–15), found 14 patients (9%) who received heart transplantation, still alive at the time of the study. Wells et al. [169] reported 11 AD-EDMD patients out of 81 patients with MDs who received HTx. However, data on age at transplant and duration of FU are not available. Four further cases of HTx in patients with EDMD are included in the group of 20 patients with MDs reported by Visrodia [192]. However, no information on these specific patients is available.

### 3.3. Limb-Girdle Muscular Dystrophies

Seventeen cases of heart transplantation in LGMDs [168,181,207,208,209,210] have so far reported. In 1998, Pitt et al. [207] described a 26-year-old woman with a diagnosis of limb-girdle muscular dystrophy and peripartum cardiomyopathy who underwent orthotopic heart transplantation. Among the five patients with muscular dystrophy who underwent heart transplantation reported by Ruiz-Cano [181], one had hips-dystrophy (LGMD) and end-stage cardiomyopathy, and was successfully transplanted at 42 years, with a FU of 6 years. Both the intraoperative and postoperative course did not show higher complication rates than in other patients.

Three cases of HTx in patients with LGMD regarded patients with LGMDR9, who share the homozygous c.826C>A, p.Leu276Ile mutation in Fukutin-related protein (*FKRP*) gene [208,209]. D’Amico et al. [208] described the first case in an 8-year-old boy presenting severe dilated cardiomyopathy with markedly elevated CK levels and no signs of muscle involvement. Margeta et al. in 2009 [209] reported two further cases: the first was a girl transplanted at 8 months for severe congestive heart failure, diagnosed at the age of 7 months. There were no significant postoperative complications, and the patient was discharged home 1 month after the surgery. At the age of 9 years, due to elevated CK values, she underwent a muscle biopsy, which allowed to establish the diagnosis of LGMD2I, by direct sequencing of the *FKRP* gene; the second case was a boy who at age 18 underwent cardiac transplantation due to worsening of heart failure. He showed, at 19 months, difficulties with walking and climbing stairs accompanied by elevated CK levels. A muscle biopsy, performed at 4 years, was abnormal, but negative for DMD. The weakness progressed slowly. At the age of 17, the patient developed shortness of breath and paroxysmal nocturnal dyspnea; echocardiogram showed severe left ventricular enlargement and reduction in left systolic function (LVEF below 20%). Muscle biopsy and direct sequencing of the *FKRP* gene, which detected also, in this case, the homozygous mutation c.826C>A, allowed for establishing the diagnosis of LGMD2I or LGMDR9. 

Of the three patients with LGMD reported in the larger NMDs series by Wu et al. in 2010 [168], as well as of the four out of 80 cases reported by Wells et al. in 2020 [169], no specific information is available on age of transplant and FU duration. Similarly, no specific information is available on the four patients with LGMD included in the group of 20 heart-transplanted individuals reported by Visrodia [192].

Kim et al. [210] recently reported a further case of heart transplant in a 39-year-old male patient with LGMD and dilated cardiomyopathy. An early cardiac rehabilitation, which included chest physiology and aerobic and resistance exercise, was performed and the patient was able to walk using a walker 28 days after HTx.

### 3.4. Myotonic Dystrophy Type 1

Patients with DM1 and concomitant cardiomyopathy are even less likely to find access to cardiac transplantation due to the perioperative risk secondary to respiratory muscle weakness. To date, only twelve of these cases have been reported in the literature. The first, dating back to 1991 [211] refers to a man with DM1 who received a cardiac allograft due to end-stage dilated cardiomyopathy. The second, described by Conraads et al. in 2002 [212], regarded the follow-up of a 40-year-old DM1 patient presenting with severe dilated cardiomyopathy and several episodes of paralytic ileus, who had a complicated post-transplant period. Of the four patients with DM1 reported by Wu et al. in 2010 [168] as well as of the two cases reported by Wells et al. in 2020 [169], no specific information is available on age of transplant and FU duration. Papa et al. [213] in 2018 described the case of a 44-year-old patient with DM1 who showed an early onset ventricular dysfunction refractory to optimal medical and cardiac resincronization therapy, and underwent successful heart transplantation. Two additional cases of HTx in patients with DM1 are included in the group of 20 patients reported by Visrodia [192], for which, however, no specific information is available. 

### 3.5. Myofibrillar Myopathies

Seven cases of heart transplantation in myofibrillar myopathies have so far been reported, all in patients with desminopathy. The first case is in Ruiz-Cano’s study [181], which evaluated the outcome of HTx in five patients with end-stage cardiomyopathy secondary to hereditary myopathies. One had desminopathy. The second case was reported by El-Menyar et al. in 2004 [214] in a member of a large Qatari family (one brother and three sisters). The brother developed restrictive cardiomyopathy at age 16. One sister underwent heart transplant for severe hypertrophic cardiomyopathy at age 15, while another sister had a permanent pacemaker for complete atrio-ventricular block at age 21. No other information is available, likely because the paper mainly focused on the discordant clinical presentation in this family. The third case, reported by Katzberg et al. in 2010 [185], regarded a 12-year-old boy with a 1-year history of exercise intolerance and shortness of breath, who was diagnosed with a restrictive cardiomyopathy and received cardiac transplant. After a few months, the patient was admitted to the intensive care unit for progressive respiratory failure due to diaphragmatic dysfunction. Muscle biopsy showed marked variation in fiber size and scattered fibers containing irregular basophilic material and amorphous eosinophilic deposits, positive on desmin immunostaining. The three patients reported by Shelly et al. in 2021 [215] were transplanted at an average age of 31.3 years (range 20–39), but nothing is said about the duration of FU. In the same year, Protonotarios et al. [166] described three patients, one of them requiring HTx, who shared the monoallelic *DES* variant c.343C>A (p.Leu115Ile), associated with malignant biventricular arrhythmogenic cardiomyopathy, a new form characterized by left ventricular dysfunction and circumferential subepicardial distribution of myocardial fibrosis.

### 3.6. Discussion

Muscular dystrophies are inherited myogenic disorders characterized by progressive muscle wasting and weakness of variable distribution and severity. They include several groups of diseases such as dystrophinopathies, Emery–Dreifuss muscular dystrophies, limb-girdle muscular-dystrophies, myotonic dystrophies, and myofibrillar myopathies. 

The genes, and their protein products that cause most of these disorders, have now been identified. This information is essential to establish an accurate diagnosis and an appropriate treatment. The heart is severely affected in several dystrophies, sometimes in the absence of clinically significant weakness. Heart transplantation can be a viable option for the treatment of a selected group of patients with end-stage heart failure muscular dystrophy associated.

The reported experience with heart transplantation, from research involving, overall, 290 patients with muscular dystrophies and end-stage dilated cardiomyopathy with refractory heart failure, is listed in Table 3, in chronological order of the studies’ publication. The analysis of the table and Figure 3 shows that the patient populations with the most access to heart transplantation are BMD (40.0%) followed by EDMD (33.8%). This is not surprising, both because cardiomyopathy is the leading cause of morbidity and mortality in these patients and because they generally have less severe skeletal and respiratory muscle dysfunction compared to patients with other forms of MDs [128,169,188]. 

Cardiomyopathy in BMD is characterized by dilation of heart chambers and systolic dysfunction leading to congestive and then refractory heart failure [3,20,27,130] while, in EDMD2, caused by mutation in *LMNA* gene, heart involvement may be more severe than in other forms, due to the presence of cardiac arrhythmias, which require the implantation of cardiac defibrillators or cardiac transplantation [50,58]. In more than 50% of cases, cardiac impairment often precedes of many years skeletal myopathy [50,58]. Cardiac surgeons often have reservations in accepting such patients, probably related to the prognosis of the underlying disease [167,175,212] or fear of possible cardiac rejection [216]. However, in recent decades, this resistance has been progressively decreasing [169,191] and, as stated by Wells et al. [169], the prognosis and life expectancy of such patients depend on the heart transplant rather than on the underlying myopathy, given that the latter can be relatively mild and without respiratory failure. Even treatment with steroids and immune-suppressants, often considered a further contraindication, can be beneficial at the muscle or cardiac level [29,30,31], or in any case, can be safe and well tolerated [217]. However, immunosuppressive therapy cannot be standardized, and must be tailored to the patient and the type of underlying myopathy. In fact, in some forms of muscular dystrophy, the massive use of immunosuppressants—often necessary in transplanted patients—could aggravate the underlying myopathy [189]. 

Among the other forms of MDs, only 3.8% of patients with DMD and 1.0% of DMD carriers have had the possibility of undergoing a heart transplant. In patients with DMD, this is probably due to the severity of the disease, leading these patients to early wheelchair use, which results in onset of scoliosis and respiratory failure, main contributing cause of death in DMD. The assessment of respiratory status in these patients is often more difficult because severe heart failure significantly affects respiratory status. However, the improvements in the treatment of respiratory complications in recent decades have reduced the mortality rate from respiratory failure, and heart failure is now the main cause of death even in this population [10]. DMD carriers still remain a neglected population from this point of view, though approximately 50% of them develop dilated cardiomyopathy and heart failure after the age of 40 [218]. Furthermore, because there are limited data available on their management, doctors often make decisions based on their own discretion.

Patients with the infantile forms of LGMDs, such as LGMD R5, R6, R11, despite presenting severe dilated cardiomyopathy with refractory heart failure, rarely have access to heart transplantation [209,210], due to the associated severe skeletal and respiratory failure. In this review, only 5.9% of patients with LGMD—a heterogeneous group of conditions with a range of phenotypes—was heart transplanted, the majority of whom had LGMDR9 (fukutin-related protein dystrophy). 

A low percentage (4.1%) of patients with DM1 and concomitant cardiomyopathy were transplanted, probably because these patients were even less likely to access cardiac transplantation due to the perioperative risk secondary to respiratory muscle weakness. 

The lowest rate of HTx was found in MFMs (2.4%), and most frequently involved patients with desminopathies, due to the high incidence of serious ventricular arrhythmias, and dilated cardiomyopathy [154,155,159,215].

The group of “unspecified type” of muscular dystrophies, which represents approximately 9% of the total HTx, included mitochondrial diseases or neuromuscular disorders, which were not the object of this review.

In all the cases described, heart transplantation appears to be an effective treatment able to prolong survival for a selected group of patients. The authors of the two largest case series of patients with MDs undergoing cardiac transplantation [168,169] agreed in stating that the intraoperative and postoperative course of these individuals did not show higher complication rates when compared to the other patients. All recipients experienced successful rehabilitation, while no evidence of graft dysfunction was observed during the follow-up. Most patients were alive, with good performance status, at the end of their studies. 

However, careful patient selection by a multidisciplinary team should take into account the severity of cardiac involvement and the prognosis related to additional skeletal muscle and respiratory symptoms, in order to determine which patients with MD-related cardiomyopathies are suitable candidates for transplantation [169,191]. Special care and consideration are also necessary during anesthesia [205], which should be adapted to prevent the development of malignant hyperthermia and rhabdomyolysis [189], and during the peri- and post-operative period to avoid life-threatening complications and limit the progression of the primary disease. Furthermore, special attention should be paid to the dosage of immune-suppressants to avoid the onset of a secondary myopathy or the occurrence of rhabdomyolysis, due to the toxic effect of cyclosporine [217], and to the adjustment of the therapy administered in order to reduce iatrogenic effects [189] and promote a faster recovery of the patients.

## 4. Conclusions

The data here reported confirm that patients with slowly progressive muscle diseases, but severe dilated cardiomyopathy and refractory heart failure, can and should undergo heart transplantation, since they seem to have a post-operative course similar to that of patients with end-stage dilated cardiomyopathy not associated with muscular diseases. Even immune-suppressive therapy, the use of which is feared due to its possible muscle-damaging effect, does not appear to affect skeletal muscles. Indeed, a study by Segushi et al. [219] shows that heart transplantation improves walking ability in patients with muscular dystrophies. Also, in our experience of four patients with end-stage cardiomyopathy secondary to dystrophin deficiency [187], heart transplantation was associated with a better prognosis, having improved muscle performance. Therefore, HTx remains the treatment of choice for patients with advanced DCM and mild-to-moderate muscle impairment. No other treatment options, such as medical therapy or electrical and/or mechanical devices, can compete with the long-term results of HTx, particularly when compared to the natural course of end-stage heart failure [3,169,191]. Even the fear of possible cardiac rejection [216] is limited—at least for patients with dystrophinopathies—because the missing/poor dystrophin present makes the production of antibodies against the deleted region of dystrophin gene very unlikely. Furthermore, the importance of rehabilitation training, both before and after heart transplantation, should be emphasized [210,212] to minimize respiratory infections and ventilatory failure in these patients. 

In conclusion, even with particular attention to the degree of skeletal myopathy, respiratory muscle involvement, and survival, it is time to overcome the residual reluctance to accept patients with muscular dystrophy-related cardiomyopathy for heart transplantation, due to the presumed reduction in life expectancy.

### Future Directions

−HTx remains the treatment of choice for patients with mild-to-moderate skeletal muscle impairment and advanced dilated cardiomyopathy.−The residual reluctance to accept patients with muscular dystrophy-related cardiomyopathy for heart transplantation due to presumed shortened life expectancy must be overcome.−Long-term prognosis in these patients is closely linked to the possibility of being transplanted.−The development of neuromuscular cardiology teams (NMCT) that include specialists in cardiology, neuromuscular diseases, pulmonology, orthopedics, endocrinology, nutrition, palliative care, and physical/occupational therapies, such as that developed by Wells et al. [169] in the USA, has proven to be quite effective in the evaluation and treatment of these patients. This, therefore, should be a good model to follow in clinical practice.−The use of correct regimens of immunosuppression therapy provides good long-term results.−Specific programs of cardiac rehabilitation are necessary, as recently reported by Kim et al. [210] and Rosenbaum et al. [220].−Guidelines, preferentially developed by the NMCT for the pre-transplant evaluation of patients, should take into account the type of myopathy, the prognosis based on the muscle disease, the cardiac and respiratory conditions of the patient, and his/her life expectancy. Furthermore, the choice of the most appropriate criteria for their selection is also advisable to optimize outcomes.−Future studies should focus on detailed outcomes to assess quality of life in these patients and better guide the NMCT in transplant decisions.

## Figures and Tables

**Figure 1 ijms-25-05289-f001:**
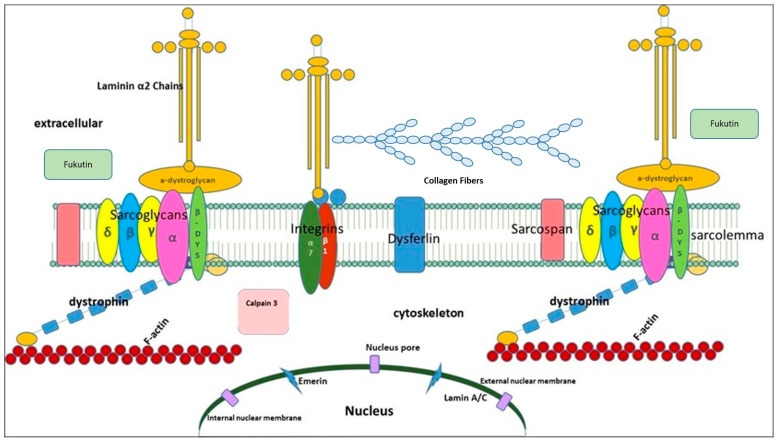
Schematic representation of proteins associated with sarcolemma in skeletal muscle fibers.

**Figure 2 ijms-25-05289-f002:**
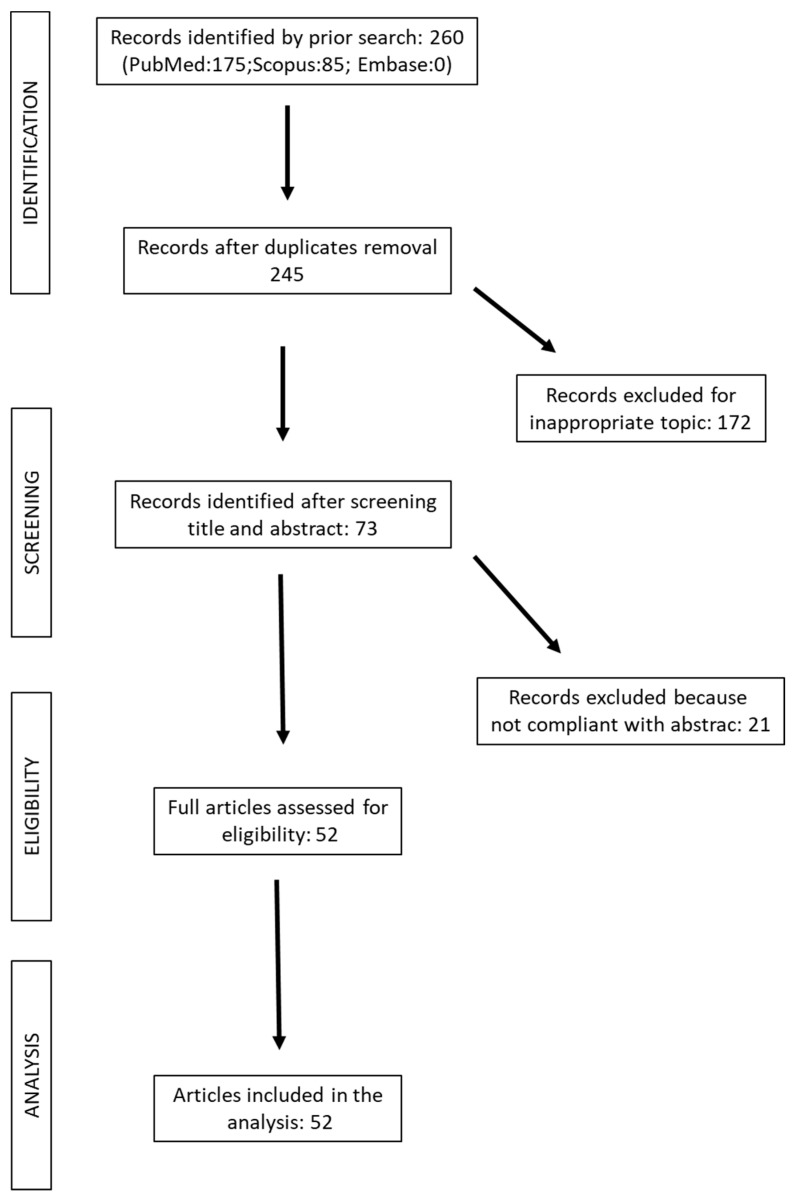
Flow-chart of the research study.

**Figure 3 ijms-25-05289-f003:**
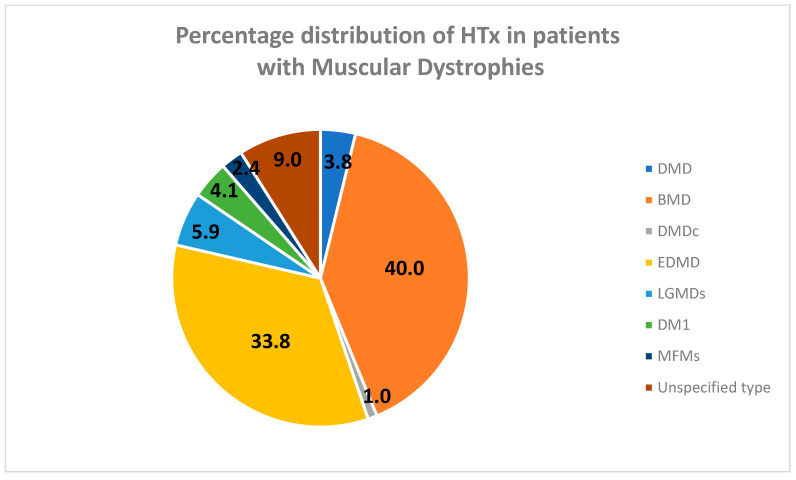
Percentage distribution of HTx in patients with muscular dystrophies.

**Table 1 ijms-25-05289-t001:** Classification of the analyzed muscular dystrophies.

Disease	Disease MIM Number	Locus	Gene Symbol	Protein
Duchenne	310200	Xp21	*DMD*	Dystrophin
Becker	300376	Xp21	*DMD*	Dystrophin
XL-DCM	302045	Xp21	*DMD*	Dystrophin
Emery-Dreifuss	310300	Xq28	*EMD*	Emerin
Myotonic Distrophy Type 1	160900	19q13	*DMPK*	Myotonin
Myofibrillar Myopathies	601419	2q35	Several	Several

Legend: MIM: Mendelian Inheritance in Man.

**Table 2 ijms-25-05289-t002:** New and old nomenclature of LGMDs according to the proposed classifications.

LGMD Type Nomenclature		MIM Disease Number	Locus	Gene Symbol	Protein
New	Old				
	1A	159000	5q31.2	*MYOT*	Myotilin
	1B	159001	1q22	*LMNA*	Lamin A/C
	1C	607801	3p25.3	*CAV3*	Caveolin-3
D1	1D	603511	7q36.3	*DNAJB6*	DNAJ/HSP40 homologue
	1E	615325	2q35	*DES*	Desmin
D2	1F	608423	7q32.1	*TPNO3*	Transportin-3
D3	1G	609115	4p21	*HNRNPDL*	Heterogeneous molecular ribonucleic D-like protein
	1H	?	?	?	Not confirmed
D4	1I	613350	15q15.1	*CAPN3*	Calpain-3
D5				*COL6A1, COL6A2, COL6A3*	Collagen 6α1, α2, α3
R1	2A	253600	15q15.1	*CAPN3*	Calpain-3
R2	2B	253601	2p13.2	*DYSF*	Dysferlin
R3	2D	608099	17q21.33	*SGCA*	α-sarcoglycan
R4	2E	604286	4q12	*SGCB*	β-sarcoglycan
R5	2C	253700	13q12.12	*SGCG*	γ-sarcoglycan
R6	2F	601287	5q33.3	*SGCD*	δ-sarcoglycan
R7	2G	601954	17q12	*TCAP*	Telethonin
R8	2H	254110	9q33.1	*TRIM32*	Tripartite motif containing protein-32
R9	2I	607155	19q13.32	*FKRP*	Fukutin-related protein
R10	2J	608807	2q32.2	*TTN*	Titin
R11	2K	609308	9q34.13	*POMT1*	Protein O-mannosyl transferase-1
R12	2L	611307	11p14.3	*ANO5*	Anoctamin-5
R13	2M	611588	9q31.2	*FKTN*	Fukutin
R14	2N	613158	14q24.3	*POMT2*	Protein O-mannosyl transferase-2
R15	2O	613157	1p34.1	*POMGnT1*	Protein O-mannose N-acetyl-glucosaminyl transferase-1
R16	2P	613818	3p21	*DAG1*	Dystroglycan
R17	2Q	613723	8q24.3	*PLEC1*	Plectin
R18	2R	615325	2q35	*DES*	Desmin
R19	2S	615356	4q35.1	*TRAPPC11*	Transpo-protein-particle-complex-11
R20	2T	615352	3p21.31	*GMPPB*	GDP-mannose-pyrophosphorylase B
R21		615618	3q13.33	*POGLUT1*	Protein O-glucosyltranferase-1
R22				*COL6A1, COL6A2, COL6A3*	Collagen 6α1, α2, α3
R23		156225	6q22.33	*LAMA2*	Laminin α2
R24		614828	3p22.1	*POMGNT2*	Protein O-linked Mannose N-acetylglucosaminyl transferase-2
R25				*BVES*	Blood vessel epicardial substance

Legend: Mendelian Inheritance in Man. “?”: unknown.

**Table 3 ijms-25-05289-t003:** Reported Experience with Heart Transplantation in patients with Muscular Dystrophies, in chronological order of publication.

Authors [Year of Publication]	DMD	BMD/ XL-DCM	DMD Carriers	EDMD	LGMDs	DM1	MFMs	Unspecified Type	Age or Mean Age at Transplant in Years (Range)	FU or Mean FU in Years (Range)	Ref. Number
Casazza et al. (1988)		1							23	2.0	[172]
D’Onofrio et al. (1989)		1							17	2.0	[173]
Sakata et al. (1990)		3							n.r.	n.r.	[174]
Merchut et al. (1990)				1					n.r.	n.r.	[196]
Anthuber et al. (1990)		1							23	2.7	[175]
Anthuber et al. (1990)				1					47	3.3	[175]
Goenen et al. (1991)						1			25	2.8	[211]
Rees et al. (1993)	3								22.3 (12–31)	2.8 (0.10–5.4)	[167]
Rees et al. (1993)		1							45	1.5	[167]
Rees et al. (1993)				1					33	2.4	[167]
Rees et al. (1993)								1	9	7.0	[167]
Piccolo et al. (1994)		1							32	n.r.	[176]
Fiocchi et al. (1997)		1							15	0.6	[177]
Pitt et al. (1998)					1				26	0.2	[207]
Melacini et al. (1998)			1						41	3.6	[193]
Finsterer et al. (1999)		1							27	6.0	[178]
Melacini et al. (2001)		1							24	0.4	[179]
Davis et al. (2001)			1						25	n.r.	[194]
Conraads et al. (2002)						1			40	5.0	[212]
Leprince et al. (2002)		1							28	1.6	[180]
Ruiz-Cano et al. (2003)		3							39.5 (24–55)	4.8 (1.1–10.8)	[181]
Ruiz-Cano et al. (2003)					1				42	6.0	[181]
Ruiz-Cano et al. (2003)							1		35	5.8	[181]
Kickuk-Chrisant et al. (2004)				2					n.r.	n.r.	[197]
El-Menyar et al. (2004)							1		n.r.	n.r.	[214]
Srinivasan et al. (2005)		1							38	n.r.	[182]
Ben Yahou et al. (2005)				2					38 (14–62)	n.r.	[198]
Komanapalli et al. (2006)		1							n.r.	n.r.	[183]
Patané et al. (2006)		1							27	1.0	[184]
Kärkkäinen et al. (2006)				6					42 (32–48)	n.r.	[199]
Cuneo et al. (2007)				1					21	n.r.	[200]
Dell’Amore et al. (2007)				2					44 (43–45)	4.5 (3.4 -5.6)	[201]
D’Amico et al. (2008)					1				8	12.0	[208]
Margeta et al. (2009)					2				0.8–18	9-n.r.	[209]
Ambrosi et al. (2009)				7					46 (21–62)	8.3 (1–17)	[202]
Wu et al. (2010)	3	15		1	3	4		3	37 (n.r.)	5.4 (n.r.—>10)	[168]
Volpi et al. (2010)				1					58	n.r.	[203]
Katzberg et al. (2010)		1							14	n.r.	[185]
Katzberg et al. (2010)							1		12	n.r.	[185]
Fournier et al. (2010)		1							53	0.3	[186]
Cripe et al. (2011)	1								14	4	[170]
Steger et al. (2013)		7							38.5 (16–56)	5.7 (1.4–11.6)	[188]
Wahbi et al. (2013)						1			45	n.r.	[100]
Blagova et al. (2016)				1					38	3.4	[204]
Papa et al. (2017)		4							30.2 (27–34)	12 (10–14.5)	[187]
Bajoras et al. (2017)		1							25	n.r.	[190]
Papa et al. (2018)						1			44	0.9	[213]
Hasselberg et al. (2019)				15					n.r.	7.8 ± 6.3	[48]
Ditaranto et al. (2019)				26					43 (34–48)	2.4 (0.8–8.5)	[50]
Peretto et al. (2019)				14					48 (35–53)	10 (7–15)	[58]
Ascencio-Lemus et al. (2019)		6							39.8 (24–48)	8.4 (0.10–19.1)	[191]
Wells et al. (2020)	3	42		11	4	2		18	n.r.	n.r.	[169]
Piperata et al. (2020)	1								18	0.3	[215]
Shelly et al. (2021)							3		31.3 (20–39)	n.r.	[166]
Protonotarios et al. (2021)							1		n.r.	n.r.	[130]
Kim et al. (2022)					1				38	n.r.	[210]
Cullom et al. (2022)			1						60	2	[195]
Visrodia et al. (2022)		6		4	4	2		4	45 (29–57.5)	5	[192]
Martins et al. (2023)				1					33	n.r.	[205]
Loureiro et al. (2023)				1					32	0.6	[206]
Total number	11	101	3	98	17	12	7	26			

Legend: n.r.: not reported.

## Data Availability

Not applicable.

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
