# Peer review of "Is Cardiac Transplantation Still a Contraindication in Patients with Muscular Dystrophy-Related End-Stage Dilated Cardiomyopathy? A Systematic Review"

_ijms, 2024, doi:10.3390/ijms25105289_

Round 1
Reviewer 1 Report
Comments and Suggestions for Authors
The systematic review on the consideration of cardiac transplantation in patients with Muscular Dystrophy-related end-stage dilated cardiomyopathy presents a well-structured and organized analysis of a complex medical issue. The clarity and coherence of the review are commendable, making it an engaging and informative read for both clinicians and researchers in the field. It is evident that considerable effort has been invested in synthesizing existing literature and providing insights into the considerations surrounding heart transplantation for this unique patient population. Overall, the systematic review appears to provide valuable insights into the consideration of cardiac transplantation in patients with Muscular Dystrophy-related end-stage dilated cardiomyopathy. However, there are some areas that need to be addressed for clarity and robustness. I have included both minor and major comments below to help strengthen the work:
MINOR COMMENTS:
1. Lines 40-42 and 127-129: merit rephrasing
2. Table 1:
a) Kindly provide full name of OMIM.
b) Also, the table lacks specific columns or indicators to differentiate between the old and new nomenclature of LGMDs, making it challenging for readers to discern which classification corresponds to each designation.
c) It appears that some LGMDs types were inadvertently omitted from then table. As per the text, there are a total of 10 dominant and 25 recessive LGMD types expected to be listed in the table. However, upon review, only 7 dominant and 23 recessive LGMD types were included.
3. Line 142: Replace EDMD2 by LGMD1 and indicate between brackets that it was replaced by EDMD2
4. Line 208: replace gene with protein
5. Figure 1: the effort put in crafting the figure illustrating the proteins associated to sarcolemma in skeletal muscle fibers is highly appreciated. However, to ensure coherence and alignment with the text, it is suggested to include the proteins present in the extracellular matrix (colla-208 gen VI, laminin 2, alpha dystroglycan, fukutin) within the figure.
6. Line 416: provide full name for FU (follow-up)
7. Line 486: HTx instead of Htx for consistency
8. Lines 525-527: is not clear and repeated in lines 559-560, lines 579-580
MAJOR COMMENTS:
1. It is essential to ensure that all claims and statements made in the systematic review are supported by appropriate references. If certain parts of the review lack references, it undermines the credibility and reliability of the findings.
e.g. Lines 258-261
Lines 334-337
Lines 350-353
Lines 357-370
Lines 450-456
Lines 467-470
Lines 634-640
Lines 638-646
2. Lines 248-251: the author adeptly identifies a significant discrepancy observed between the findings of the study Kaspar et al. and the study of Restrepo-Cordoba et al. regarding the correlation between the type of mutations and the age of onset of dilated cardiomyopathy in patients with Muscular Dystrophy-related conditions. This highlights a critical area of contention and underscores the need for further investigation and clarification. While the observation itself is valuable, providing analysis and interpretation could enhance the review's depth and utility. By delving into potential explanations for this inconsistency, such as methodological differences or sample size limitations, the author could offer valuable insights that guide future research directions and clinical practice. Addressing this gap would not only strengthen the review's scholarly rigor but also contribute to a more comprehensive understanding of the complex interplay between genetic factors and disease progression in this patient population.
3. While the author concludes that heart transplantation remains the treatment of choice for these patients, it is crucial to acknowledge any limitations or uncertainties in the existing evidence. Are there any conflicting findings or ongoing debates in the literature that should be mentioned?
4. While the author mentions high operative risk and potential graft involvement as reasons for reluctance toward cardiac transplantation, it would be beneficial to provide more specific data or studies to support these claims. Without clear evidence, these assertions may appear speculative.
5. The suggested future directions are pertinent and could significantly enhance the management of Muscular Dystrophy-related cardiomyopathy. However, it would be beneficial to elaborate on how these recommendations could be implemented in clinical practice.
a) For example, regarding the recommendation for multidisciplinary care, are there specific models of care coordination that have been successful in similar patient populations?
b) While the review mentions the importance of correct regimens of immunosuppression therapy, it might be beneficial to provide more details on what constitutes "correct regimens" in this context. Are there any specific considerations or adjustments that need to be made for patients with Muscular Dystrophy-related cardiomyopathy compared to other transplant recipients?
c) The suggestion for developing guidelines for the pre-transplant evaluation of these patients is crucial. However, it would be helpful to outline what factors should be included in these guidelines, how they might differ from standard evaluation criteria for heart transplantation candidates, and how they can be implemented effectively.
Comments on the Quality of English Language
The quality of English is generally commendable. The writing is clear, coherent and easily understood. However, few phrases, as indicated in the comments section, may benefit from rephrasing to enhance clarity and precision.
Author Response
Responses to Reviewers
I want to thanks the reviewers for their positive evaluation of my manuscript and for their constructive and appropriate comments/suggestions that helped me to improve it.
Below, the point-by-point answers to their questions.
Reviewer 1
The systematic review on the consideration of cardiac transplantation in patients with Muscular Dystrophy-related end-stage dilated cardiomyopathy presents a well-structured and organized analysis of a complex medical issue. The clarity and coherence of the review are commendable, making it an engaging and informative read for both clinicians and researchers in the field. It is evident that considerable effort has been invested in synthesizing existing literature and providing insights into the considerations surrounding heart transplantation for this unique patient population. Overall, the systematic review appears to provide valuable insights into the consideration of cardiac transplantation in patients with Muscular Dystrophy-related end-stage dilated cardiomyopathy. However, there are some areas that need to be addressed for clarity and robustness. I have included both minor and major comments below to help strengthen the work:
MINOR COMMENTS:
- Lines 40-42 and 127-129: merit rephrasing
Answer: Following your suggestion, the sentence has been rephrased as follows: “Cardiac and respiratory complications are the most frequent cause of death in patients with NMDs. However, while significant progress in the treatment of respiratory failure has led to increased survival and improved quality of life in these patients over the past several decades [6–8], cardiac involvement still remains a major cause of morbidity and mortality [9–11].
- Table 1:
- a)Kindly provide full name of OMIM.
Answer: Done (Online Mendelian Inheritance in Man)
- b)Also, the table lacks specific columns or indicators to differentiate between the old and new nomenclature of LGMDs, making it challenging for readers to discern which classification corresponds to each designation.
Answer: I have redrawn the table (now number 2) to make it clearer
- c)It appears that some LGMDs types were inadvertently omitted from then table. As per the text, there are a total of 10 dominant and 25 recessive LGMD types expected to be listed in the table. However, upon review, only 7 dominant and 23 recessive LGMD types were included.
Answer: See my answer to point b.
Line 142: Replace EDMD2 by LGMD1 and indicate between brackets that it was replaced by EDMD2
Answer: Done (Now, line 152).
- Line 208: replace gene with protein
Answer: Done
- Figure 1: the effort put in crafting the figure illustrating the proteins associated to sarcolemma in skeletal muscle fibers is highly appreciated. However, to ensure coherence and alignment with the text, it is suggested to include the proteins present in the extracellular matrix (collagen VI, laminin 2, alpha dystroglycan, fukutin) within the figure.
Answer: Thank you for appreciating the effort made to draw the figure. As suggested, the two missing elements (collagen VI and fukutin) have now been added in the extracellular space.
Line 416: provide full name for FU (follow-up)
Answer: Done
- Line 486: HTx instead of Htx for consistency
Answer: Thanks for pointing out the typo. I corrected it.
- Lines 525-527: is not clear and repeated in lines 559-560, lines 579-580
Answer: Heart transplants in subjects affected by Muscular Dystrophies are often included in larger series of patients in which data are provided in an aggregate manner,and not reported for each individual form of disease. Therefore, it was impossible to indicate age of transplant and duration of follow-up for these specific patients. This applies to patients reported in lines 525-527, lines 559-560, lines 579-580.
However, for greater clarity, I modified the sentences as follows:
- (lines 525-527, now lines 573-577 ): “A further case of HTx in a patient with EDMD was reported by Wu et al. in 2010 [160, now 168]; however, no information about age of transplant and duration of follow-up of this specific patient is available”
- (lines 559-560, now lines 606-608): “Of the three patients with LGMD reported in a larger NMDs series by Wu et al. in 2010 [160, now 168] as well as of the four out of 80 cases reported by Wells et al. in 2020 [161, now 169], no specific information is available on age of transplant and FU duration
- (lines 579-580, now 611-613):” Similarly, no specific information is available on the four patients with LGMD included in the group of 20 heart transplanted individuals with NMDs reported by Visrodia [184, now 192]”
MAJOR COMMENTS:
- It is essential to ensure that all claims and statements made in the systematic review are supported by appropriate references. If certain parts of the review lack references, it undermines the credibility and reliability of the findings.
e.g. Lines 258-261
Lines 334-337
Lines 350-353
Lines 357-370
Lines 450-456
Lines 467-470
Lines 634-640
Lines 638-646
Answer: The appropriate references were reported at lines indicated.
- Lines 248-251: the author adeptly identifies a significant discrepancy observed between the findings of the study Kaspar et al. and the study of Restrepo-Cordoba et al. regarding the correlation between the type of mutations and the age of onset of dilated cardiomyopathy in patients with Muscular Dystrophy-related conditions. This highlights a critical area of contention and underscores the need for further investigation and clarification. While the observation itself is valuable, providing analysis and interpretation could enhance the review's depth and utility. By delving into potential explanations for this inconsistency, such as methodological differences or sample size limitations, the author could offer valuable insights that guide future research directions and clinical practice. Addressing this gap would not only strengthen the review's scholarly rigor but also contribute to a more comprehensive understanding of the complex interplay between genetic factors and disease progression in this patient population.
Answer: Following your suggestion, I better clarified this concept as follows (now, lines 247-276):
“In 2009, Kaspar et al. [120, now 128] analysed 78 patients with BMD and XL-DCM with common deletions predicted to alter the dystrophin protein, and correlated these mutations to the age of onset of cardiomyopathy. They found that deletions affecting the amino-terminal domain (deletions of exons 2-9 of the DMD gene) are associated with early-onset DCM (mid-20′s), while deletions removing part of the rod domain and hinge 3 (deletions of exons 45-49 of the DMD gene) have a later onset DCM (mid-40′s). By combining genetic and protein information, their analysis revealed a strong correlation between specific protein structural modifications and age of onset of dilated cardiomyopathy. In particular, they found that cardiac dystrophin may be particularly sensitive to structural disruptions of the exon 45-49 region compared to skeletal muscle dystrophin. Their conclusions were in agreement with the results of the studies in dystrophin-null mdx mice expressing a mini-dystrophin construct that lacks the exon 45 to 49 region, but has an intact hinge 3 domain. In these mice, only a partial restoration of cardiac function was achieved in spite of a complete rescue of the skeletal muscle pathology [121, now 129]. However, Restrepo-Cordoba et al. [122, now 130] in their series of 112 patients with DCM associated to DMD gene mutations with and without muscle impairment, have recently shown no relation between type of mutation and DCM and no difference between type of mutation (truncating and non-truncating variants) [130] and the mean age at DCM diagnosis. However, it should be noted that the two papers cannot be compared, because they consider two different patient populations and have different objectives. Kaspar et al. [128] studied a population of patients with Becker phenotype and analyzed in detail the association type of deletion-onset of cardiomyopathy. In contrast, Restrepo-Cordoba et al. [130] analyzed a population of patients with cardiac phenotype (dilated cardiomyopathy) associated with mutations in the DMD gene. Moreover, among the 79 patients who also had skeletal myopathy, only six shared the deletion 45-49, considered crucial by Kaspar et al. Furthermore, the Restrepo-Cordoba’s study aimed to describe the prognosis of dystrophin-associated DCM in patients without skeletal myopathy, underlining how these patients should be offered lifelong surveillance to diagnose and manage cardiac complications”.
- While the author concludes that heart transplantation remains the treatment of choice for these patients, it is crucial to acknowledge any limitations or uncertainties in the existing evidence. Are there any conflicting findings or ongoing debates in the literature that should be mentioned?
Answer: The sentence was reworded as follows (lines 687-696): Cardiac surgeons often had reservations in accepting such patients, probably related to the prognosis of the underlying disease [159, 167, 204] or fear of cardiac rejection [208]. However, in recent decades, this resistance has been progressively decreasing [169,191] and, as stated by Wells et al. [169] prognosis and life expectancy of such patients depend on the heart transplant rather than the underlying myopathy, given that the latter can be relatively mild and without respiratory failure. Even treatment with steroids and immune-suppressants [217], often considered a further contra-indication, may be beneficial at the muscle and cardiac level [29-32] or in any case be safe and well tolerated [217].
- While the author mentions high operative risk and potential graft involvement as reasons for reluctance toward cardiac transplantation, it would be beneficial to provide more specific data or studies to support these claims. Without clear evidence, these assertions may appear speculative.
Answer: See answer to point 3.
- The suggested future directions are pertinent and could significantly enhance the management of Muscular Dystrophy-related cardiomyopathy. However, it would be beneficial to elaborate on how these recommendations could be implemented in clinical practice.
- a)For example, regarding the recommendation for multidisciplinary care, are there specific models of care coordination that have been successful in similar patient populations?
Answer: The sentence was changed as follows (lines 778-782): “The development of neuromuscular cardiology teams that include specialists in cardiology, neuromuscular diseases, pulmonology, orthopedics, endocrinology, nutrition, palliative care, and physical/occupational therapies, such as that developed by Wells et al. [169] in the USA, has proven to be quite effective in the evaluation and treatment of these patients. This therefore should be a model to follow in clinical practice.”
- b)While the review mentions the importance of correct regimens of immunosuppression therapy, it might be beneficial to provide more details on what constitutes "correct regimens" in this context. Are there any specific considerations or adjustments that need to be made for patients with Muscular Dystrophy-related cardiomyopathy compared to other transplant recipients?
Answer: The following sentence was added (lines 696-700):”However, immunosuppressant therapy cannot be standardized, and should be tailored to the patient and the type of underlying myopathy. In fact, in some forms of muscular dystrophy the massive use of immunosuppressants - often necessary in transplanted patients - could aggravate the underlying myopathy [189]”.
- c)The suggestion for developing guidelines for the pre-transplant evaluation of these patients is crucial. However, it would be helpful to outline what factors should be included in these guidelines, how they might differ from standard evaluation criteria for heart transplantation candidates, and how they can be implemented effectively.
Answer: The sentence was modified as follows (lines 794-798): “Guidelines, preferentially developed by the neuromuscular cardiology team for the pre-transplant evaluation of patients, should take into account the type of myopathy, the prognosis based on the muscle disease, the cardiac and respiratory conditions of the patient and his/her life expectancy. Furthermore the choice of the most appropriate criteria for their selection is also advisable to optimize outcomes”.
Comments on the Quality of English Language
The quality of English is generally commendable. The writing is clear, coherent and easily understood. However, few phrases, as indicated in the comments section, may benefit from rephrasing to enhance clarity and precision.

Reviewer 2 Report
Comments and Suggestions for Authors
This is a review by Luisa Politano titled “Is cardiac transplantation still a contraindication in patients with Muscular Dystrophy-related end-stage dilated cardiomyopathy? A systematic review” where the author has reported an overview of cardiac involvement in MDs and its possible association with the underlying molecular defects, as well as a systematic review of Heart transplant (HTx) outcomes in patients with MD- related end-stage dilated cardiomyopathy, published so far in the literature. Cardiac muscle can be variably affected my MD that depends on the genetic basis of the MD. Onset and progression of cardiac involvement are independent of the development of skeletal myopathy and a Poor correlation exists between genotypes and phenotypes at the cardiac and skeletal muscle levels. Moreover, patients with inherited muscle disorders and end-stage cardiomyopathies, HTx is still considered a relative contraindication. Therefore, the area of this review is of high significance to the field of research.
Overall, the review was written very well and presented in a systemic manner. The discussions and the conclusion section are broadly described. The future perspective is helpful. A few minor comments are listed below that will help improve the quality of this paper.
1. A list of abbreviations will be helpful as several keywords are used.
2. The resolution of Figure 2 is poor. One can hardly read the text. Please enhance the quality of this figure.
3. The author describes LGMD in detail, supported by a table describing the old and new nomenclature. Why is there no information on the other type of MD?
4. A table describing various genes responsible for each type of MDs which have reported evidence of cardiovascular abnormalities would be helpful.
5. Please discuss briefly on the aspects of genders (male vs female). This will give an insight into the heterogeneity in terms of gender.
Author Response
I want to thanks the reviewers for their positive evaluation of my manuscript and for their constructive and appropriate comments/suggestions that helped me to improve it.
Below, the point-by-point answers to their questions.
Response to the Reviewer 2
This is a review by Luisa Politano titled “Is cardiac transplantation still a contraindication in patients with Muscular Dystrophy-related end-stage dilated cardiomyopathy? A systematic review” where the author has reported an overview of cardiac involvement in MDs and its possible association with the underlying molecular defects, as well as a systematic review of Heart transplant (HTx) outcomes in patients with MD- related end-stage dilated cardiomyopathy, published so far in the literature. Cardiac muscle can be variably affected my MD that depends on the genetic basis of the MD. Onset and progression of cardiac involvement are independent of the development of skeletal myopathy and a Poor correlation exists between genotypes and phenotypes at the cardiac and skeletal muscle levels. Moreover, patients with inherited muscle disorders and end-stage cardiomyopathies, HTx is still considered a relative contraindication. Therefore, the area of this review is of high significance to the field of research.
Overall, the review was written very well and presented in a systemic manner. The discussions and the conclusion section are broadly described. The future perspective is helpful. A few minor comments are listed below that will help improve the quality of this paper.
- A list of abbreviations will be helpful as several keywords are used.
Answer: A list of abbreviations and acronyms has been added to the end of the manuscript.
- The resolution of Figure 2 is poor. One can hardly read the text. Please enhance the quality of this figure.
Answer: Done
- The author describes LGMD in detail, supported by a table describing the old and new nomenclature. Why is there no information on the other type of MD?
Answer: I did not include any information on these types of MDs because they are well defined, at both clinical and genetic level. By contrast, the group of LGMD is more complex and needed ,in my opinion, a particular attention. In any case, as suggested, a new table (Table 1) has been added for the other types of MDs included in this review.
- A table describing various genes responsible for each type of MDs which have reported evidence of cardiovascular abnormalities would be helpful.
Answer: See answer to point 3.
- Please discuss briefly on the aspects of genders (male vs female). This will give an insight into the heterogeneity in terms of gender.
Answer: Few information regarding this particular aspect is available in the literature. However, I added a new brief paragraph (2.6, lines 204-223) to discuss these aspects, according to your request:
“Gender differences regarding muscle involvement are reported in numerous limb-girdle dystrophies. For example, in those due to defects in Calpain 3 [116], anoctamine 5 [117,118] and teletonin [119], males are more severely affected than females, while in those caused by defects in dysferlin [120] or FKRP [121] females are more affected than males. On the contrary, cardiac aspects have rarely been investigated and little information is available in the literature probably because the most frequent and severe forms of muscular dystrophy such as DMD, BMD and EDMD1 usually affect males. Only two papers deal with this topic in the literature. The first [122] reports that male patients with R9 Limb-Girdle Muscular Dystrophy, homozygous for the c.826C>A mutation, have a positive correlation with the onset of cardiomyopathy but not with age or stage of the disease. The second [123] which concerns patients with sarcoglycanopathies reports no correlation with sex but only with the duration of the disease”

Reviewer 3 Report
Comments and Suggestions for Authors
Hereditary muscular dystrophies are associated with cardiac involvement to varying degrees depending on the genetic basis. Politano's review provides compelling evidence that patients with the most severe forms of dilated cardiomyopathy and/or conduction system abnormalities in slowly progressive muscle disease should undergo heart transplantation. The well-structured, comprehensive review needs minor modifications.
1. Please give a putative explanation for the divergent results reported in references 120 and 122.
2. Please give a putative explanation for the contradictory results reported in references 141 and 142.
3. Please indicate which desmin mutation (p.N116S vs. p.Glu401Asp) is associated with right or left ventricular arrhythmogenic cardiomyopathy.
4. Line 395 (page 10): “dilated heart transplantation”
Is the Author referring to a heart transplantation due to DCM?
Comments on the Quality of English LanguageMinor editing of English language is suggested.
Author Response
I want to thanks the reviewers for their positive evaluation of my manuscript and for their constructive and appropriate comments/suggestions that helped me to improve it.
Below, the point-by-point answers to their questions.
Reviewer 3
Hereditary muscular dystrophies are associated with cardiac involvement to varying degrees depending on the genetic basis. Politano's review provides compelling evidence that patients with the most severe forms of dilated cardiomyopathy and/or conduction system abnormalities in slowly progressive muscle disease should undergo heart transplantation. The well-structured, comprehensive review needs minor modifications.
- Please give a putative explanation for the divergent results reported in references 120 and 122.
Answer: To explain the apparent divergent results, the sentence was rephrased as follows:
In 2009, Kaspar et al. [120, now 128] analysed 78 patients with BMD and XL-DCM with common deletions predicted to alter the dystrophin protein, and correlated these mutations to the age of onset of cardiomyopathy. They found that deletions affecting the amino-terminal domain (deletions of exons 2-9 of the DMD gene) are associated with early-onset DCM (mid-20′s), while deletions removing part of the rod domain and hinge 3 (deletions of exons 45-49 of the DMD gene) have a later onset DCM (mid-40′s). By combining genetic and protein information, their analysis revealed a strong correlation between specific protein structural modifications and age of onset of dilated cardiomyopathy. In particular, they found that cardiac dystrophin may be particularly sensitive to structural disruptions of the exon 45-49 region compared to skeletal muscle dystrophin. Their conclusions were in agreement with the results of the studies in dystrophin-null mdx mice expressing a mini-dystrophin construct that lacks the exon 45 to 49 region, but has an intact hinge 3 domain. In these mice, only a partial restoration of cardiac function was achieved in spite of a complete rescue of the skeletal muscle pathology [121, now 129]. However, Restrepo-Cordoba et al.[122, now 130] in their series of 112 patients with DCM associated to DMD gene mutations, with and without muscle impairment, have recently shown no relation of the type of mutation with DCM and no difference between the type of mutations (truncating and non-truncating variants [130] and the mean age at DCM diagnosis. However, it should be noted that the two papers cannot be compared because they consider two different patient populations and have different objectives. Kaspar et al. [129] studied a population of patients with Becker phenotype and analyzed in detail the association type of deletion-onset of cardiomyopathy. In contrast, Restrepo-Cordoba et al. [130] analyzed a population of patients with cardiac phenotype (dilated cardiomyopathy) associated with mutations in the DMD gene. Moreover, among the 79 patients who also had skeletal myopathy, only six shared the deletion 45-49, considered crucial by Kaspar et al. Furthermore, the Restrepo-Cordoba’s study aimed to describe the prognosis of dystrophin-associated DCM in patients without skeletal myopathy, underlining how these patients should be offered lifelong surveillance to diagnose and manage cardiac complications”.
- Please give a putative explanation for the contradictory results reported in references 141 and 142 (now 69 and 149).
Answer: I rephrased the sentence as follows (lines 354-358) “The discordant results between the studies could depend on the number of patients investigated, as well as on the prevalence of the homozygosity condition in some countries or on having included other types of mutations in the heterozygous group”.
- Please indicate which desmin mutation (p.N116S vs. p.Glu401Asp) is associated with right or left ventricular arrhythmogenic cardiomyopathy.
Answer: The association is now better indicated in the main text: “Desmin mutation pN116S is associated with right ventricular arrhythmogenic cardiomyopathy (lines 418-420), while desmin mutation p.Glu401Asp is associated with both left and right ventricular arrhythmogenic cardiomyopathy (lines 425-427)”.
- Line 395 (page 10): “dilated heart transplantation”
Is the Author referring to a heart transplantation due to DCM?
Answer: Yes, correct.
Comments on the Quality of English Language
Minor editing of English language is suggested

Round 2
Reviewer 1 Report
Comments and Suggestions for Authors
The reviewer would like to thank the author for their efforts in addressing the comments provided on their systematic review. The revised manuscript has been reviewed, and the attention the author has given to most of the suggested revisions is highly appreciated. While many of the previous comments have been addressed satisfactorily, there are still a couple of points that require further attention.
Firstly, it appears that some references are still missing for certain statements, as previously indicated.
e.g. Lines 258-261
Lines 334-337
Lines 467-470
Secondly, the table comparing the old and new nomenclature of LGMDs still lacks clarity in delineating between the two classifications. Incorporating distinct columns or labels to indicate the old and new classifications for each LGMD subtype would greatly enhance the table's utility and facilitate understanding for readers.
Thirdly, Line 159: it is LGMD1 and not LGMD2
Additionally, a few new minor comments have arisen upon reevaluation. Please find below:
Line 753: adjust space between words and letters
Line 268 and line 281: remove now from brackets
Line 759: new paragraph for “Among….”
